# Genetic Engineering of Lesquerella with Increased Ricinoleic Acid Content in Seed Oil

**DOI:** 10.3390/plants10061093

**Published:** 2021-05-29

**Authors:** Grace Q. Chen, Kumiko Johnson, Tara J. Nazarenus, Grisel Ponciano, Eva Morales, Edgar B. Cahoon

**Affiliations:** 1Western Regional Research Center, Agricultural Research Service, U.S. Department of Agriculture, 800 Buchanan St., Albany, CA 94710, USA; kumiko.johnson@usda.gov (K.J.); Grisel.ponciano@usda.gov (G.P.); evadesalta@gmail.com (E.M.); 2Department of Biochemistry and Center for Plant Science Innovation, University of Nebraska-Lincoln, Lincoln, NE 68588, USA; tnazarenus2@unl.edu (T.J.N.); ecahoon2@unl.edu (E.B.C.)

**Keywords:** hydroxy fatty acid, ricinoleic acid, lesquerolic acid, triacylglycerol, *Physaria fendleri*, lesquerella, seed oil, RNA interference, genetic transformation

## Abstract

Seeds of castor (*Ricinus communis*) are enriched in oil with high levels of the industrially valuable fatty acid ricinoleic acid (18:1OH), but production of this plant is limited because of the cooccurrence of the ricin toxin in its seeds. Lesquerella (*Physaria fendleri*) is being developed as an alternative industrial oilseed because its seeds accumulate lesquerolic acid (20:1OH), an elongated form of 18:1OH in seed oil which lacks toxins. Synthesis of 20:1OH is through elongation of 18:1OH by a lesquerella elongase, PfKCS18. Oleic acid (18:1) is the substrate for 18:1OH synthesis, but it is also used by fatty acid desaturase 2 (FAD2) and FAD3 to sequentially produce linoleic and linolenic acids. To develop lesquerella that produces 18:1OH-rich seed oils such as castor, RNA interference sequences targeting *KCS18*, *FAD2* and *FAD3* were introduced to lesquerella to suppress the elongation and desaturation steps. Seeds from transgenic lines had increased 18:1OH to 1.1–26.6% compared with that of 0.4–0.6% in wild-type (WT) seeds. Multiple lines had reduced 18:1OH levels in the T_2_ generation, including a top line with 18:1OH reduced from 26.7% to 19%. Transgenic lines also accumulated more 18:1 than that of WT, indicating that 18:1 is not efficiently used for 18:1OH synthesis and accumulation. Factors limiting 18:1OH accumulation and new targets for further increasing 18:1OH production are discussed. Our results provide insights into complex mechanisms of oil biosynthesis in lesquerella and show the biotechnological potential to tailor lesquerella seeds to produce castor-like industrial oil functionality.

## 1. Introduction

The conventional source of hydroxy fatty acid (HFA) is castor (*Ricinus communis*), which contains 90% ricinoleic acid (18:1OH) in its seed oil. The fatty acid and its derivatives are used as feedstocks for numerous industrial products, such as lubricants, plastics and surfactants [1,2]. The production of castor oil is hampered by the presence of the toxin ricin [3,4] and hyper-allergenic 2S albumins [5,6,7] in its seeds. Lesquerella (*Physaria fendleri*, Brassicaceae) seed oil contains a major HFA, lesquerolic acid (20:1OH) at 55–60% [8,9,10,11], and seeds of this plant lack any known toxins. As such, efforts have been made through plant breeding to develop lesquerella as a new oilseed crop that is a safe source of HFA [12,13]. With the success of lesquerella biotechnology [14,15] and the deep knowledge of genes for fatty acid and seed oil biosynthesis [16,17,18], lesquerella oil can be improved through metabolic engineering [15].

Seed oil (triacylglycerol, TAG) starts from de novo fatty acid (FA) biosynthesis in plastid and TAG assembly in endoplasmic reticulum (ER) [16]. Simplified pathways and genes in lesquerella TAG synthesis are shown in Figure 1.

During lesquerella seed development, oleic acid (18:1) is synthesized in plastid, exported and activated to 18:1-Coenzyme A (CoA) in the cytosol. The 18:1-CoA can be acylated directly into membrane lipid phosphatidylcholine (PC) in the ER by the forwarding reaction of lyso-PC acyltransferase (LPCAT) [19,20,21] resulting in 18:1-PC (Figure 1). The 18:1-PC is the substrate of oleate 12-hydroxylase (FAH12) [22,23,24,25] which hydroxylates 18:1-PC to form 18:1OH-PC (Figure 1). Lesquerella PfFAH12 is bi-functional FAD2-related oleate Δ12-hydroxylase:desaturase that converts 18:1-PC to both 18:1OH-PC and linoleic acid (18:2)-PC [25]. Through the reverse reaction of LPCAT (Figure 1), or phospholipase A (PLA2)–type activity [26], the 18:1OH can be removed from PC, following its synthesis on this lipid, and transferred back to cytosol to be activated as 18:1OH-CoA. A lesquerella seed fatty acid condensing enzyme (PfKCS18) (also known as KCS3 or FAE1, all designations used in this article) elongates 18:1OH-CoA to 20:1OH-CoA [27] (Figure 1). Rapid acylation and de-acylation by LPCAT (or by PLA2), and in conjunction with efficient elongation by PfKCS18 leads to enrichment of 20:1OH-CoA in cytosol. PC is also the substrate for FA desaturase 2 (FAD2) [28] and FA desaturase 3 (FAD3) [29] that sequentially converts 18:1 to 18:2 and 18:2 to linolenic acids (18:3), respectively (Figure 1). In addition to 20:1OH, 18:1OH, 18:1, 18:2 and 18:3, lesquerella oil contains auricolic acid (20:2OH), also formed by FAD3 [30,31]. Lesquerella PfFAD3-1 is a key enzyme producing 18:3 and 20:2OH [32]. FA-CoA in cytosol or FA-PC in ER are assembled to TAG through multiple mechanisms [16,19] (Li 2013; Bates 2016). Kennedy pathway [33] is the major route for FA esterification into TAG, which consists of three sequential acylations of FA-CoAs to a glycerol-3-phosphate (G3P) backbone. The *sn*-1 position of G3P is acylated by glycerol-3-phosphate acyltransferase (GPAT) to produce lysophosphatidic acid (LPA). The *sn*-2 position of LPA is acylated by LPA acyltransferase (LPAT) to generate phosphatidic acid (PA). PA is then converted to 1,2-*sn*-diacylglycerol (DAG, or de novo DAG) by PA phosphatase (PAP). Finally, the *sn*-3 position of DAG is acylated by 1,2-*sn*-diacylglycerol acyltransferase (DGAT) to produce TAG. Lesquerella TAGs contain ~60% 20:1OH, and almost all of 20:1OH are acylated to the *sn*-1 and *sn*-3 positions, and the *sn*-2 positions of lesquerella TAGs are exclusively occupied by unsaturated FAs, i.e., 18:1, 18:2 and 18:3 [34,35,36,37]. The reason for lack of HFA at the *sn*-2 position of TAG has been suggested, in part, by the selectivity of lesquerella LPAT (PfLPAT) for unsaturated FA [15], which is a common feature for most plant microsomal LPAT [38]. PC can be converted to DAG (PC-derived DAG) through the removal of the head group from the PC by PC:DAG cholinephosphotransferase (PDCT) [39,40,41] (Figure 1); therefore, acyl-CoAs on the PC are directed to DAG for TAG synthesis. PC-derived DAG can be produced by the reverse action of CDP-choline: DAG cholinephosphotransferase (CPT) [42], a lipase-based mechanism using phospholipase C (PLC), or phospholipase D plus PAP [19,43]. Because FAs in *sn*-2-PC can be modified, (e.g., desaturation and hydroxylation), the conversion of PC into DAG also provides a means to increase the amount of modified FAs (mFAs) such as 18:2, 18:3 and 18:1OH, in *sn*-2-TAG. Moreover, FA on the *sn*-2 PC can be transferred to the *sn*-3 position of DAG by phospholipid:DAG acyltransferase (PDAT) [44,45,46] (Figure 1).

Castor genes are introduced into non-HFA-oilseed arabidopsis (*Arabidopsis thaliana*) or camelina (*Camelina Sativa*) to study the HFA biosynthesis mechanism [47]. Castor *RcFAH12* was first isolated and demonstrated to be responsible for HFAs synthesis in transgenic seeds up to 17% [24,48,49]. Additional genes, castor *RcDGAT2* [50], *RcPDAT1-2* (or *RcPDAT1-A*) [45,46], and *RcPDCT* [41], *RcPLCL1* [43], *RcLPAT2* [51,52], *RcLPAT3B* and *RcLPATB* [52], and *RcGPAT9* together with *RcLPAT2* and *RcPDAT1A* [53], are demonstrated to increase the HFAs content of transgenic arabidopsis or camelina from 17 to 28%. 

One of our research goals is to generate a castor-oil producing lesquerella that is safe, cost-competitive, and widely accepted as an industrial feedstock. We have previously attempted to generate a castor oil-producing lesquerella through over-expressing of a castor *RcLPAT2* involved in TAG assembly. In that study we demonstrated that seed oils of transgenic lesquerella showed increases in 18:1OH from 1% to 4%, and castor oil-like TAGs from 5% to 14% [15]. In this study, we aimed to further enhance the 18:1OH level by down-regulating the expression of lesquerella *PfFAD2, PfFAD3*, and *PfKCS18* genes using RNA interference (RNAi) technology [54,55]. A RNAi silencing approach was used to suppress an endogenous target gene expression through transgenic expression of a double-stranded RNA (dsRNA) that shares sequence homology with the target and leads to cleavage of the targeted transcripts, [54,55]. We hypothesize that suppressing *FAD2* and *FAD3* reduces polyunsaturated FAs (PUFAs) levels, including 18:2 and 18:3, and subsequently increases 18:1, the substrate of PfFAH12 (Figure 1). Suppressing *KCS18* reduces 20:1OH and subsequently increases 18:1OH for TAG assembly. PfFAD2, PfFAD3, and PfKCS18 share high sequence homology with camelina CsFAD2, CsFAD3, and arabidopsis AtKCS18, showing 93.6%, 95.6%, and 82.3%, respectively [18]. RNAi constructs, *CsFAD2 RNAi*, *CsFAD3 RNAi* and *AtFEA1 RNAi,* are effective in silencing corresponding gene expression in camelina [56,57,58]. We therefore generated transgenic lesquerella expressing *CsFAD2 RNAi*, *CsFAD3 RNAi* and *AtFEA1 RNAi*. We have demonstrated here that high levels of 18:1OH can be achieved by blocking the desaturation and elongation steps. Our results not only provide tools for engineering castor oil-producing lesquerella, but also enhance our understanding of the mechanisms of HFA synthesis.

## 2. Results

### 2.1. Changes of FA Composition in Transgenic Lesquerella Expressing Two dsRNAs, AtFAD3 RNAi + CsFAE1 RNAi

We produced 16 independent lines expressing *AtFAD3 RNAi* + *CsFAE1 RNAi* (2-dsRNA) under the control of seed-specific glycinin promoters. Mendelian segregation analysis on Basta resistance of T_1_ seeds revealed that five lines had one transgenic locus, 10 lines had two loci, and one had more than two loci (Appendix A). Variable FA compositions were observed among these 16 lines. To aid examination, line 1 to line 16 were assigned to these transgenics based on descending order in 18:1OH content in their seed oils (Table 1). Moderate positive correlation (*r* = 0.45, *p* = 0.08) was observed between 18:1OH content and transgenic copy number. In all seeds, five minor fatty acids, palmitic (16:0), palmitoleic (16:1), stearic (18:0), arachidic (20:0), and eicosenoic (20:1) acids, had very low levels and small changes from 1.6–2.3%, 0.3–0.8%, 0.9–1.8%, 0.1–0.3%, 0.4–0.8%, respectively, (Appendix A). When these minor FAs were combined, there were no significant differences between each transgenic line and wild-type (WT) (Table 1). Consistent with our hypothesis, we observed significant increases in 18:1OH content among 13 lines ranging from 1.2% (line 13) to 26.6% (line 1) compared with 0.6% of WT (Table 1) and decreases in 20:1OH among lines 1–10 ranging from 19% (line 1) to 46.9% (line10) compared with 51.2% of WT (Table 1). Except for line 13, 18:3 was significantly reduced in all lines ranging from to 1.5% (line 14) to 9.6% (line 10) compared with 13.3% of WT. All transgenic lines increased in 18:2, ranging from 9.5% (line 13) to 20% (line 14) compared with 7.6% of WT (Table 1). Notably, 18:1 content was increased in line 1–7 and line 10, ranging from 19% (line 10) to 32.1% (line 2) compared with 17% of WT (Table 1). In all transgenic lines, 20:2OH content was reduced significantly ranging from 0–2.2% compared with 4.3% of WT (Table 1). Total HFA content dropped from 56% (WT) to 42.9–53.8% among lines 1–10 and increased slightly to 57.7–57.8% in line 15 and line 16 (Table 1).

In greenhouse-grown plants we produced selfed T_2_ seeds by hand-pollination of individual Basta-resistant T_1_ plants. Multiple T_2_ seed population lines were obtained from each line 1 to line 6 as they contained higher levels of 18:1OH than the remaining lines (Table 1). Results of FA analysis for each T_2_ line is shown in Figure 2. As in the T_1_ generation, the five minor FAs combined did not show significant changes between WT and each T_2_ transgenic line (Figure 2, Appendix A). However, we did not find any T_2_ off-springs containing 18:1OH at a level higher than their parents. T_2_ populations from line 1 and line 2 had substantial reduction in 18:1OH content, from 26.6% to 19–9% (line 1-1–1-9) and 16.8% to 7.3–2.3% (line 2-1–2-4), respectively, (Figure 2). For line 3 to line 6, similar levels of 18:1OH were maintain in their top T_2_ off-springs, showing 14.6% (line 3-1), 10.6% (line 4-1), 9.8% (line 5-1), and 6.8% (line 6-1) compared with their T_1_ parents at 16.6%, 11.7%, 10.2% and 8.5%, respectively, (Figure 2); low levels of 18:1OH were observed in line 3-8, line 4-9, and line 5-8 and line 6-9 at 2.6%, 3.7%, 1.6% and 2.9%, respectively, (Figure 2).

### 2.2. Changes of FA Composition in Transgenic Lesquerella Expressing Three dsRNAs, CsFAD2 RNAi + AtFAD3 RNAi + CsFAE1 RNAi

Fifteen independent transgenic lines expressing the 3-dsRNAs, *CsFAD2 RNAi* + *AtFAD3 RNAi* + *CsFAE1 RNAi* were generated and their T_1_ seeds were analyzed for FA composition. Once again, the five minor fatty acids, palmitic (16:0), palmitoleic (16:1), stearic (18:0), arachidic (20:0), and eicosenoic (20:1) acids, had slight variation among transgenic lines (Appendix A). There was no significant difference on the total minor FAs content between each transgenic line and WT (Table 2). For the other FAs, we observed similar average contents in 18:2, 18:3 and 20:1 between the group expressing 2-dsRNA (*AtFAD3 RNAi* + *CsFAE1 RNAi*) (Table 1) and the group expressing 3-dsRNA (Table 2). Noticeably, the 3-dsRNA group with the addition of *CsFAD2 RNAi* accumulated more 18:1 at the average of 27.8% (Table 2) compared with the average of 20.5% in the 2-dsRNA group (Table 1 and Table 2). In addition, the increase in average 18:1OH and decrease in average total HFA were less dynamic in the 3-dsRNA group, showing averages of 4.7% and 48.9%, respectively, (Table 2), compared with that of 7.7% and 53% in lines expressing 2-dsRNA, respectively, (Table 1 and Table 2). The fatty acid composition of WT presented in Table 1 and Table 2, and Figure 2 are similar to previously described [11]. We did not observe any changes of growth phenotype for all transgenic lesquerella lines.

### 2.3. Correlations between FA Levels among Transgenic Lines

Correlation analysis was performed to show the relationships between FA accumulation for 2-dsRNA group at T_1_ and T_2_ generations, and 3-dsRNA group (Appendix A). As expected for the impact of *CsFAE1 RNAi*, strong negative correlations were displayed between 18:1OH and 20:1OH (−0.99 ≤ *r* ≥ −0.75) in all groups examined (Appendix A). Similarly, for the impact of *AtFAD3 RNAi*, strong negative correlations between 18:2 and 18:3 (−0.92 ≤ *r* ≥ −0.67) were also shown in all groups examined (Appendix A). The impact of *CsFAD2 RNAi* in 3-dsRNA lines exhibited weak negative correlation between 18:1 and 18:2 (*r* = −0.38) and very weak positive correlation between 18:1 and 18:1OH (*r* = 0.15) (Appendix A). For both 2-dsRNA and 3-dsRNA groups, we observed strong negative correlations between total HFA and 18:1 (−0.82 ≤ *r* ≥ −0.92), and strong positive correlations between total HFA and 20:1OH (0.75 ≤ *r* ≥ 0.96) (Appendix A).

## 3. Discussion

### 3.1. High Levels of 18:1OH Accumulate in Lesquerella by Blocking Elongation and Desaturation of Fatty Acids

In this study, the dsRNA fragments in *AtFAD3 RNAi* and *CsFAE1 RNAi* contain a 323 bp or 251 bp sequence sharing 91.7% and 82.8% identity with lesquerella *PfFAD3-1* (BenBank ID: MF611845) [32] and *PfKCS18* (GenBank ID: AF367052) [27], respectively. When the 2-dsRNAs (*AtFAD3 RNAi* and *CsFAE1 RNAi*) were introduced to lesquerella, we observed changes in FA composition (Table 1). Among the 16 T_1_ transgenic lesquerella lines, 15 lines shifted the accumulation of 18:3 to 18:2, showing a strong negative correlation between 18:2 and 18:3 (*r* = 0.93) (Appendix A); 13 lines shifted 20:1OH to 18:1OH, which also displayed a strong negative correlation (*r* = −0.99) (Appendix A). These results indicate that *AtFAD3 RNAi* and *CsFAE1 RNAi* are effective in silencing *PfFDA3-1* and *PfKCS18*, respectively. To see the effect of *AtFAD3 RNAi* and *CsFAE1 RNAi* in the next generation, we examined FA composition in T_2_ seeds from the top six T_1_ lines. The line 1 seeds produced the highest 18:1OH content at 26.6% in the T_1_ generation, however, the 18:1OH decreased from 19% (line 1-1) to 9% (line 1-9) in the T_2_ generation (Figure 2). Similar significant reduction of 18:1OH also occurred in line 2 from 16.8% (T_1_) to 7.3% (T_2_ line 2-1) to 2.3% (T_2_ line 2-4). The remaining top best T_2_ off-springs from line 3 to line 6 also showed reductions in 18:1OH contents, but the reductions were not as large, varying between line 3 from 11.7% (T_1_) to 10.57% (T_2_ line 3-1) and line 6 from 8.5% (T_1_) to 6.8% (T_2_ line 6-1). The reduction of 18:1OH content occurring in all 6 top T_2_ lines implies that the competence of *CsFAE1 RNAi* was not fully transmitted to the next generation. A similar phenomenon was reported for an *RNAi* in arabidopsis where the influence of the silencing faded through several selfed generations due to a generation-dependent decrease in transcription of the RNAi [59]. In maize, analysis of an RNAi effect over multiple generations also reveals that some lines display reduced transgene silencing, but the effect of the RNAi can be maintained by outcrossing rather than self-pollination [60]. Such phenomenon is explained based on the assumption that hemizygosity would reduce any potential *trans*-interactions between the transgenes on homologous chromosomes that could lead to transgene silencing [60]. The molecular basis of *CsFAE1 RNAi* stability in lesquerella remains to be investigated. Another possibility is that the 18:1OH contents in these transgenic lesquerella lines resulted from equilibrium of FA and TAG metabolism, and there could be a ceiling for 18:1OH accumulation in lesquerella. We generated 31 independent lines expressing *CsFAE1 RNAi* (Table 1 and Table 2), only one line accumulated a high level of 18:1OH at 26.6% and it dropped to 19% in the next generation (Table 1). The second highest three lines contain 18:1OH ranging from 15–17% (Table 1 and Table 2). Thus, a highest stable equilibrium of 18:1OH level could fall between 15–20%. These lines are useful for further assessment of 18:1OH accumulation limits and relationships between *CsFAE1 RNAi* effect and 18:1OH levels in lesquerella. Assuming the silencing effects of *CsFAE1 RNAi* and *AtFAD3 RNAi* led to increases in 18:1OH and decreases in 18:3, respectively, there were 24 out of 31 (77%) of the transgenic lines with increased 18:1OH levels, whereas 30 out of 31 (97%) of the lines showed decreased 18:3 content (Table 1 and Table 2). The results indicate that the silencing effect of *AtFAD3 RNAi* is more stable than that of *CsFAE1 RNAi*, which could be attributed to the higher nucleotide identity of 91.7% displayed between *AtFAD3 RNAi* and *PfFAD3-1* than the 82.8% identity demonstrated between *CsFAE1 RNAi* and *PfKCS18.*

In a separate experiment, we introduced a construct carrying three dsRNAs, *AtFAD3 RNAi* + *CsFAE1 RNAi* + *CsFAD2 RNAi* into lesquerella and generated 15 independent transgenic lines. We observed strong negative correlation between 18:2 and 18:3 (*r* = −0.67), and between 18:1OH and 20:1OH (*r* = −0.75), indicating the strong impacts of *AtFAD3 RNAi* and *CsFAE1 RNAi* shown again in transgenics expressing the 3-dsRNAs construct (Table 2 and Appendix A). Regarding the effect of *CsFAD2 RNAi*, we observed a weak negative correlation between 18:1 and 18:2 (*r* = −0.38), but this is opposite to the weak positive correlation between 18:1 and 18:2 (*r* = 0.38) observed in the 2-dsRNAs which did not contain *CsFAD2 RNAi* (Appendix A). The result suggests that *CsFAD2 RNAi* exerts certain silencing effect which resulted in shifting the accumulation of 18:2 to 18:1 in lesquerella. Unlike *PfFAD3-1* and *PfKCS18* which were specifically targeted by *AtFAD3 RNAi* and *CsFAE1 RNAi*, respectively, the *CsFAD2 RNAi* may target two homologous lesquerella genes, *PfFAD2* (GenBank ID: DQ518313) and *PfFAH12* (GenBank ID: KC972619) that share 78.3% nucleotide identity. In fact, the dsRNA fragment in *CsFAD2 RNAi* contains 299 bp sequences which exhibit 55.7% and 88.9% identity with *PfFAD2* and *PfFAH12*, respectively. The effect of *CsFAD2 RNAi* on silencing *PfFAH12* can be inferred by the reduction of correlation strength between 18:1 and 18:1OH from very strong positive (*r* = 0.93) in 2-dsRNAs lines, which did not express *CsFAD2 RNAi* (Table 1 and Appendix A), to very weak positive (*r* = 0.15) in 3-dsRNA lines due to *CsFAD2 RNAi* (Table 2 and Appendix A). The shift of 18:1OH accumulation to 18:1 was also evident by lower accumulation of 18:1OH at an average of 4.4% in the 3-dsRNAs lines compared with that of 7.7% (average) in the 2-dsRNAs lines (Table 1 and Table 2). The silencing effect of *CsFAD2 RNAi* on both *PfFAD2* and *PfFAH12* gene expression can be deduced by the increased accumulation of 18:1 content at an average of 27.8% in lines expressing 3-dsRNAs (Table 2) compared with that of 20.5% in lines expressing 2-dsRNAs (Table 1). Our results support previous observations of using these *RNAi* sequences to generate high 18:1 content in camelina lines, including *CsFAD2 RNAi* [56], *CsFAD2 RNAi* + *CsFAE1 RNAi* [57], or *CsFAD2 RNAi + AtFAD3 RNAi + CsFAE1 RNAi* [58] (in preparation).

### 3.2. Constrains and Potential for Production of a High 18:1OH-Containing Oil in Lesquerella

Expression of *CsFAE1 RNAi, CsFAD2 RNAi*, and *AtFAD3 RNAi* in lesquerella resulted in significant increases not only in 18:1OH, but also in 18:1 (Table 1 and Table 2). The results indicated that 18:1 was inefficiently used for synthesizing 18:1OH. This could be partially due to the nature of PfFAH12 which is a bifunctional oleate hydroxylase:desaturase [25], that may not efficiently convert 18:1 to 18:1OH. Seed oil from castor or *Physaria lindheimeri* contains 90% 18:1OH [61] or 85% 20:1OH [62], respectively. These species have distinct *FAH12*s, *RcFAH12* in castor [24] and *PlFAH12* in *P. lindheimeri* [62]. Replacement of *PfFHA12* with *RcFAH12* or *PlFAH12* should allow more efficient 18:1OH synthesis in lesquerella. Alternatively, the resulted substantial accumulation of 18:1 could also be due to some lesquerella endogenous genes having substrate preference to 18:1 and efficiently incorporating 18:1 into TAG. During seed development, a lesquerella LPAT acts like a typical plant LPAT that has substrate preference for unsaturated FAs including18:1-CoA, resulting in efficient incorporation of 18:1-CoA into TAG through the Kennedy pathway. Castor *RcLPAT2* is useful for increasing 18:1OH at the *sn*-2 of TAGs in lesquerella [15,37]. Additional isoforms, *RcLPAT3B* and *RcLPATB*, have also been shown to increase 18:1OH in arabidopsis seed TAGs [52]. Substituting the lesquerella endogenous *PfLPAT* with these specific castor *RcLPATs* may increase 18:1OH flux to TAG by *RcLPATs*. Besides the Kennedy pathway, PC-derived DAG pathway may also channel 18:1 into TAG by a lesquerella PfPDCT (Figure 1). Once 18:1-PC is synthesized, e.g., by PfLPCAT, some of the 18:1-PC could be converted by PDCT to 18:1-DAG for TAG assembly (Figure 1). Lesquerella seed TAGs contain about 21% PUFAs (18:2 and 18:3) (Table 1 and Table 2). There is strong evidence that plants enriched with PUFAs in seed TAG may use the PC-derived pathway [19]. Therefore, it is likely that PC-derived DAGs are utilized in TAG assembly in lesquerella. Castor gene *RcPDCT* was demonstrated to enhance flux from 18:1OH-PC to 18:1OH-DAG [41]. It would be favorable to over-express *RcPDCT* in lesquerella to increase 18:1OH incorporation to TAG through PC-derived DAG pathway. 

It is anticipated that the increased 18:1OH is at the expense of 20:1OH in transgenic lesquerella lines expressing *CsFAE1 RNAi* (Table 1 and Table 2), however, total HFA decreased and showed strong correlation with 20:1OH contents (0.96 < *r* > 0.75, Appendix A). The results indicated that 18:1OH was not incorporated into TAG at the same efficiency as 20:1OH. Lesquerella PfKCS18 is evolved to specifically elongate 18:1OH-CoA to 20:1OH-CoA [27] (Figure 1). It is possible that other lesquerella enzymes, such as PfGPAT, PfDGAT and/or PfPDAT also co-evolved to adapt and utilize 20:1OH efficiently. Most plant GPATs have a broad acyl-CoA substrate specificity [19,63]. There is evidence that castor RcGPAT9 plays an important role in acylating HFAs at the *sn*-1 position of G3P, resulting in *sn*-1-HFA-LPA, which facilitates the subsequent incorporation of *sn*-2 and *sn*-3 HFA into seed TAG by LPAT and DGAT [53]. In plant seeds accumulating unusual FAs, members of DGAT2 family are essential enzymes in acylating unusual FAs to the s*n*-3 position of DAG. For example, castor RcDGAT2 prefers 18:1OH to common FAs [50,64]. Lesquerella seed transcriptome analysis reveals one PfGPAT9 and three PfDGATs [17]. It would be interesting to explore whether these genes have substrate selectivity for HFA-CoA or common FA-CoA. The role of PfKCS18 has been explored in camelina [65]. Transgenic camelina expressing RcFAH12 accumulates 15% HFA [66] but the resulted transgenic seeds reduce TAG content and seed germination ability [65]. When *RcFAH12* with *PfKCS18* are co-expressed, the transgenic camelina seeds increase HFA content to 21% and also restore TAG content and seed germination ability [65]. Camelina is not a native species for HFA synthesis, 18:1OH-PC generated by RcFAH12 in camelina may be subjected to β-oxidation [67], or represents a bottleneck [40], limiting HFA accumulation. The elongation step by PfKCS18 may ease the 18:1OH flux from PC to cytosol FA-CoA pool, thus relieve the bottleneck and facilitate the utilization of HFA-CoA by the Kennedy pathway [33] (Figure 1). PDAT transfer FA at the *sn*-2 position of PC to the *sn*-3 position of DAG, yielding TAG [44,68] (Figure 1). Castor has two *PDAT1s*, but only *RcPDAT1-2* (or *RcPDAT1A*) selects 18:1OH-PC as substrate, and it participates in HFA-TAG synthesis [45,46]. There are three PfPDATs expressed in lesquerella seeds [17]. Whether these PfPDATs are involved in transferring HFA-PC to DAG remains to be investigated. To further enhance 18:1OH accumulation in lesquerella TAGs, coordinated expression of multiple genes, such as *RcGPAT9*, *RcDGAT2* and *RcPDAT1-2* (or *RcPDAT1A*) should promote 18:1OH accumulation in seed TAG. 

In summary, to develop a castor oil-producing lesquerella crop, we designed genetic engineering schemes based on known pathways of fatty acid biosynthesis in lesquerella. As predicted, high levels of 18:1OH were accumulated by reducing the elongation of 18:1OH to 20:1OH through expression of *CsFAE1 RNAi*. Additionally, high levels of 18:1 and 18:2 were accumulated through suppression of desaturation steps by expressing *CsFAD2 RNAi* and/or *AtFAD3 RNAi*. Intriguingly, the accumulated 18:1 was not efficiently utilized to produce 18:1OH and instead, 18:1 was largely channeled to seed TAG. On the other hand as discussed, multiple mechanisms could limit the acylation of 18:1OH into TAG. Our results direct future research efforts in implementing genetic approach that targets not only enhancement of 18:1OH synthesis, but also on increased 18:1OH acylation to TAG. Nevertheless, we demonstrated for the first time that lesquerella can be engineered for large increases in 18:1OH levels from 0.4–0.5% in WT to a stable high level of 15–20% in transgenic seed oils. 

## 4. Materials and Methods

### 4.1. Construction of pBinGlyBar1 + AtFAD3 RNAi + CsFAE1 RNAi and pBinGlyBar1 CsFAD2 RNAi + AtFAD3 RNAi + CsFAE1RNAi 

Constructs used for transformation experiments were prepared as follows. The *FAD3 RNAi* hairpin cassette was prepared by PCR amplification of a 323-bp fragment of the FAD3 gene from *Arabidopsis thaliana* Col-0 cDNA in both antisense using primers Arm1-5′*Nhe*I-F3 5′- AATAA*GCTAGC*ACCGGACACACCACCAGAAC-3′ and Arm1-3′EcoRI-F3 5′- TATT*GAATTC*CGTAGACTTTAAGAACCGCGAG-3′ and sense orientations using primers Arm2-5′*Pst*I-F3 5′- TAATA*CTGCAG*CACCGGACACACCACCAGAAC-3′ and Arm2-3′*Xho*I-F3 5′- ATTA*CTCGAG*CCGTAGACTTTAAGAACCGCGAG-3′ and cloned into plasmid pGEMT-Easy-HTM3 [69], replacing the existing antisense and sense arms. The resulting *FAD3* hairpin sequence was excised at *Eco*RI/*Xho*I from that plasmid and inserted into pBinGlyBar1 [59] as an *Eco*RI/*Xho*I fragment. Flanking the *FAD3* hairpin sequence in pBinGlyBar1 was the seed-specific promoter and the 3′UTR for the *Glycine max* glycinin-1 gene. The new construct was designated *pBinGlyBar1 + AtFAD3 RNAi*. The *FAE1* RNAi suppression cassette was prepared by PCR amplification of a 251-bp portion of the camelina *FAE1* gene from cDNA using the oligonucleotides: 5′-TAAT*TCTAGACTCGAG*GGGAATACTTCGTCTAGCTC-3′ and 5′-TATA*AAGCTTACTAGT*CCGACCGTTTTTTGACATGAGTC-3′. The PCR product was assembled sequentially in an inverted repeat orientation of either side of the *Flaveria trinervia* pyruvate orthophosphate dikinase (Pdk) intron [70]. The hairpin cassette was then cloned downstream of the seed-specific promoter for the *Glycine max* glycinin-1 gene and upstream of the 3′UTR for the glycinin-1 gene as a *Not*1 fragment. The resulting vector contained *Asc*I restriction sites that flanked the glycinin-1 promoter and 3′UTR. Using this restriction site the entire cassette containing promoter, RNAi hairpin and 3′UTR were assembled into the *Asc*I site of the binary vector pBinGlyBar1 + AtFAD3 RNAi. The resulting construct was designated pBinGlyBar1 + AtFAD3 RNAi + cFAE1 RNAi. The *FAD2* RNAi hairpin cassette was prepared by PCR amplification of a 299–bp portion of the camelina *FAD2* gene from cDNA using the oligonucleotides:5′-TAAT*TCTAGACTCGAG*CGTCTTGATCACTTACTTGCAG-3′ and 5′-TATA*AAGCTTACTAGT*CTACATAGATACACTCCTTTGCC-3′. The product was cloned sequentially in an inverted repeat orientation of either side of the *Flaveria trinervia* pyruvate orthophosphate dikinase (Pdk) intron. The hairpin cassette was then cloned downstream of the seed-specific promoter for the soybean oleosin gene and upstream of the oleosin 3′UTR as a *Not*1 fragment. The resulting vector contained *Asc*I restriction sites that flanked the oleosin promoter and 3′UTR. Using this restriction site the entire cassette containing promoter, RNAi hairpin and 3′UTR were assembled into the *Mlu*I site of *pBinGlyBar1 + AtFAD3 RNAi + cFAE1 RNAi* to make *pBinGlyBar1 + AtFAD3 RNAi + cFAE1 RNAi + cFAD2 RNAi* which also contains a *bar* marker gene for Basta selection of transgenic plants.

### 4.2. Plant Transformation and Growth Condition

The lesquerella seeds, WCL-LY2 [71] were kindly provided by Dave Dierig (USDA-ARS, Arid-Land Agricultural Research Center, Maricopa, AZ, USA). Plant transformation was performed using the Agrobacterium tumefaciens strain AGL1 [72] carrying the binary vector *pBinGlyBar1 + AtFAD3 RNAi + CsFAE1 RNAi and pBinGlyBar1 CsFAD2 RNAi + AtFAD3 RNAi + CsFAE1RNAi*. Tissue culture and plant growth conditions were as described before [14] with the exception of using Basta (1 mg/L) as a transgenic selective agent. In brief, leaves harvested from plants in sterile condition were wounded by slightly scratching the underside of the leaf and then dipping the leaf in the half strength MS medium containing the Agrobacterium for 5 min. Following the inoculation, leaves were blotted on sterilized filter paper and transferred to Callus and Shoot Induction (CSI) medium composed of basal medium (BM, half strength MS medium plus 30 g L^−1^ sucrose and 6 g L^−1^ agar, pH 5.7) supplemented with 1 mg L^−1^ 6-benzylaminopurine (BA) and 0.1 mg L^−1^ 1-Naphthaleneacetic Acid (NAA). After incubating the infected leaves in the growth chamber for 2 days, the leaves were cut into 5 mm segments and cultured on CSI media plus 1 mg L^−1^ Basta for transgenic selection and 100 mg L^−1^ timentin for inhibiting the Agrobaterium growth. In 6–8 weeks, yellow-greenish Basta resistant calli started to appear on the leaf segments. To eliminate chimeras, each shoot was cut into small pieces (about 2 × 2 mm^2^) and placed on the CSI medium for shoot regeneration. After 4 rounds of successive regenerations, shoots were sub-cultured on BM plus 1 mg L^−1^ BA, 1 mg L^−1^ Indole-3-Butyric Acid (IBA) and 1 mg L^−1^ Basta for multiplication. Shoots 10–15 mm in length were transferred to rooting medium (BM plus 1 mg L^−1^ IBA and 50 mg L^−1^ Basta). When a shoot developed 2–3 roots (usually in 3–5 weeks), it was then transferred to a Magenta box (Sigma, St. Louis, Mo) containing sterilized peat-vermiculite growth mixture (Sunshine mix #4, Planet Natural, Bozeman, MT) presoaked with 1 mg L^−1^ IBA water solution. After 8–10 weeks in the growth mixture, well-developed primary plants showing 8–12 normal leaves and 2–3 inch height were transferred to a 6-inch pot and placed under a transparent plastic cover for the first 2 weeks for acclimation in the greenhouse. T_1_ selfed seeds were obtained by hand-pollination between different flowers from the same transgenic plant (T_0_). To estimate the number of transgene locus for each line, T_1_ seeds were germinated on germination medium containing Basta at 1 mg L^−1^ for 3 weeks. Healthy seedlings showing normally developed cotyledons and 2–4 true leaves were counted as resistance seedlings (R); sensitive seedlings (S) had arrested yellow cotyledons and no true leaves. Transgene locus numbers were based on the Mendelian ratio of R:S, 3:1 for one locus, 16:1 for two loci. T_1_ seedlings were transplanted into soil for T_2_ seed production.

### 4.3. Analysis of Fatty Acid Composition

Seeds homogenized in a gas chromatograph (GC) autosampler vial were subjected to direct transesterification to produce fatty acid methyl esters (FAMEs) using trimethyl-sulphonium hydroxide (TMSH) as described [59,73]. The resulting FAMES were analyzed by GC-flame ionization detection using previously described instrument conditions [74].

## Figures and Tables

**Figure 1 plants-10-01093-f001:**
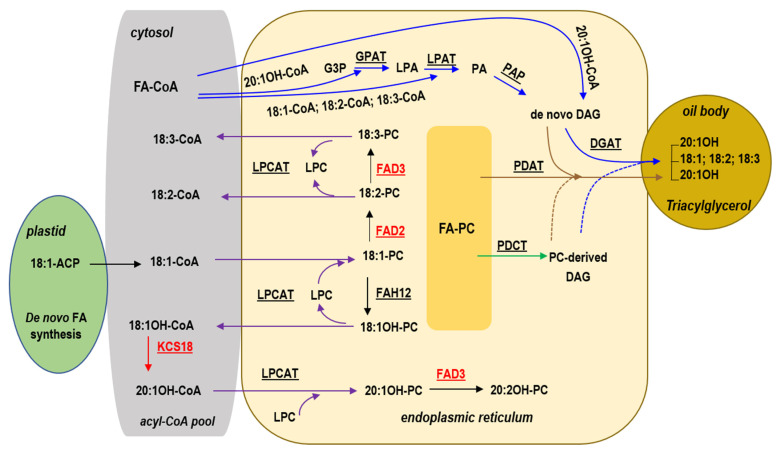
Simplified pathways for fatty acid and triacylglycerol synthesis in lesquerella seeds. Blue arrows indicate reactions involved in the Kennedy pathway. Purple arrows indicate reactions involved in acyl editing by LPCAT. Brown arrows indicate PDAT-mediated pathways. Green arrow indicates reactions involved in PC-derived DAG synthesis. Dotted lines indicate PC-derived DAG utilized by DGAT and PDAT. The red arrow indicates the elongation step by KCS18. Enzymes catalyzing these reactions are underlined. Red fonts are the targeted enzymes in this study. Fatty acid numerical symbols: 18:1, oleic acid; 18:1OH, ricinoleic acid; 20:1OH, lesquerolic acid; 20:2OH, auricolic acid; 18:2, linoleic acid; 18:3, linolenic acid. Abbreviations: CoA, co-enzyme A; PC, phosphatidylcholine; LPCAT, lysophosphatidylcholine acyltransferase; LPC, lysophosphatidylcholine; FAH12, Δ12 oleic acid hydroxylase; KCS18, 3-ketoacyl-CoA synthase 18; G3P, glycerol-3-phosphate; LPA, lysophosphatidic acid; PA, phosphatidic acid; DAG, diacylglycerol; GPAT, glycerol 3-phosphate acyltransferase; LPAT, lysophosphatidic acid acyltransferase; PAP, phosphatidic acid phosphatase; DGAT, diacylglycerol acyltransferase; PDAT, phospholipid:DAG acyltransferase; PDCT, PC:DAG cholinephosphotransferase; TAG, triacylglycerol.

**Figure 2 plants-10-01093-f002:**
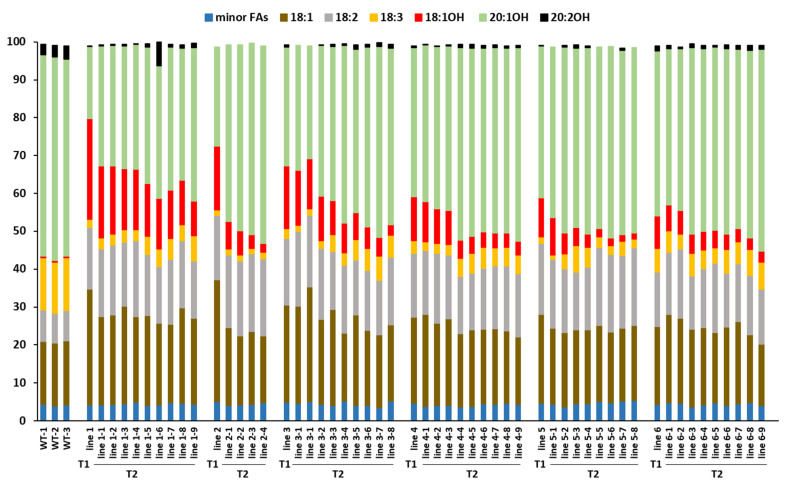
Fatty acid content in T2 seeds expressing AtFAD3 RNAi + CsFAE1 RNAi. Triplicates of 30-seed samples were measured for wild-type (WT) and each transgenic line. Fatty acid legend: 18:1 is oleic; 18:2 is linoleic; 18:3 is linolenic; 18:1OH is ricinoleic; 20:1OH is lesquerolic; and 20:2OH is auricolic acid. Total content of five common minor fatty acids: palmitic (16:0), palmitoleic (16:1), stearic (18:0), Arachidic acid (20:0), and eicosenoic acid (20:2OH).

**Table 1 plants-10-01093-t001:** Fatty acid composition (mole %) in T1 seeds expressing *AtFAD3 RNAi* + *CsFAE1 RNAi.*

Line	Total Minor Fatty Acid ^a^	18:1	18:2	18:3	18:1OH	20:1OH	20:2OH	Total Hydroxy Fatty Acid
wild-type	4.7 ± 0.4	17.0 ± 0.4	7.6 ± 0.4	13.3 ± 0.6	0.6 ± 0.2	51.2 ± 1.0	4.3 ± 0.6	56.0 ± 0.5
line 1	4.3 ± 0.2	30.5 ± 2.8 ***	16.2 ± 1.7 ***	2.2 ± 0.7 ***	26.6 ± 0.2 ***	19.0 ± 2.0 ***	0.2 ± 0.2 ***	45.8 ± 1.9 ***
line 2	5.1 ± 0.2	32.1 ± 1.3 **	17.0 ± 0.5 ***	1.7 ± 0.5 ***	16.8 ± 0.5 ***	26.1 ± 1.0 ***	0.0 ± 0.0 ***	42.9 ± 0.5 ***
line 3	5.0 ± 0.2	25.5 ± 1.9 **	17.6 ± 1.2 ***	2.6 ± 1.4 ***	16.6 ± 1.0 ***	31.4 ± 2.1 ***	0.5 ± 0.4 ***	48.4 ± 1.5 ***
line 4	4.7 ± 0.4	22.7 ± 2.7 *	16.8 ± 0.6 ***	3.3 ± 0.8 ***	11.7 ± 2.7 **	39.3 ± 5.0 **	0.4 ± 0.2 ***	51.4 ± 2.6 *
line 5	4.7 ± 0.2	23.4 ± 2.0 **	18.7 ± 0.2 ***	1.8 ± 0.6 ***	10.2 ± 0.4 ***	40.2 ± 1.4 ***	0.1 ± 0.1 ***	50.5 ± 1.5 ***
line 6	4.4 ± 0.1	20.5 ± 1.8 *	14.4 ±1.0 ***	6.2 ± 0.5 ***	8.5 ± 0.4 ***	43.7 ± 1.4 **	1.2 ± 0.1 ***	53.4 ± 1.5 *
line 7	4.1 ± 0.1	20.5 ± 0.9 **	16.7 ± 2.0 ***	4.9 ± 1.7 ***	8.0 ± 1.3 ***	45.5 ± 2.0 **	0.3 ± 0.2 ***	53.8 ± 1.2 *
line 8	4.5 ± 0.0	17.6 ± 0.5	16.1 ± 0.5 ***	7.6 ± 0.4 ***	7.5 ± 0.5 ***	45.0 ± 0.5 ***	1.3 ± 0.2 ***	53.8 ± 0.2 **
line 9	4.4 ± 0.1	17.9 ± 0.5	17.2 ± 0.7 ***	4.3 ± 0.7 ***	4.9 ± 0.4 ***	49.3 ± 0.6 *	0.7 ± 0.2 ***	54.9 ± 0.5
line 10	4.8 ± 0.0	19.0 ± 1.0 *	13.0 ± 0.5 ***	9.6 ± 0.5 ***	4.7 ± 1.2 **	46.9 ± 0.6 *	1.2 ± 0.1 ***	52.8 ± 1.4 *
line 11	4.3 ± 0.0	16.5 ± 0.4	16.3 ± 0.2 ***	5.4 ± 0.3 ***	3.6 ± 0.3 ***	51.9 ± 0.4	0.8 ± 0.2 ***	56.2 ± 0.0
line 12	4.1 ± 0.2	16.3 ± 0.5	14.1 ± 0.9 ***	7.7 ± 1.2 **	1.8 ± 0.3 **	53.2 ± 1.0	1.8 ± 0.2 **	56.7 ± 1.1
line 13	3.9 ± 0.3	17.1 ± 2.1	9.5 ± 0.2 **	13.3 ± 0.4	1.2 ± 0.1 **	51.1 ± 2.7	2.2 ± 0.1 **	54.5 ± 2.7
line 14	4.3 ± 0.3	16.2 ± 0.5	20.0 ± 0.7 ***	1.5 ± 0.1 ***	0.5 ± 0.1	56.2 ± 1.2 **	0.1 ± 0.0 ***	56.8 ± 1.3
line 15	4.0 ± 0.4	16.1 ± 0.5	14.2 ± 1.2 ***	6.6 ± 0.9 ***	0.5 ± 0.1	55.5 ± 0.7 **	1.8 ± 0.4 **	57.7 ± 0.8 *
line 16	4.1 ± 0.1	15.4 ± 0.4 **	13.4 ± 1.5 **	8.3 ± 1.3 **	0.5 ± 0.1	55.6 ± 0.2 **	1.7 ± 0.3 **	57.8 ± 0.5 *
average of transgenics	4.4 ± 0.1	20.5 ± 0.9	15.7 ± 0.5	5.4 ± 0.4	7.7 ± 0.7	44.4 ± 1.2	0.9 ± 0.1	53.0 ± 0.8

Three or four replicates of 30-seed samples were measured for wild-type and each transgenic line. All data are averages of measurements ±SD. Fatty acid legend: 18:1 is oleic; 18:2 is linoleic; 18:3 is linolenic; 18:1OH is ricinoleic; 20:1OH is lesquerolic; and 20:2OH is auricolic acid. ^a^, total content of five common fatty acids: palmitic (16:0), palmitoleic (16:1), stearic (18:0), arachidic (20:0), and eicosenoic acids (20:1). Two-tailed Student’s *t*-test. * *p* < 0.05; ** *p* < 0.01; *** *p* < 0.001.

**Table 2 plants-10-01093-t002:** Fatty acid composition (mole %) in T1 seeds expressing CsFAD2 RNAi + AtFAD3 RNAi + CsFAE1 RNAi.

Line	Total Minor Fatty Acid ^a^	18:1	18:2	18:3	18:1OH	20:1OH	20:2OH	Total Hydroxy Fatty Acid
wild-type	4.2 ± 0.2	16.7 ± 0.2	8.0 ± 0.2	13.7 ± 0.2	0.40 ± 0.0	53.0 ± 0.9	3.1 ± 0.5	56.5 ± 0.7
line 1	4.2 ± 0.1	27.7 ± 0.3 ***	13.8 ± 0.4 ***	4.8 ± 0.5 ***	15.4 ± 0.7 ***	33.3 ± 0.8 ***	0.9 ± 0.1 ***	49.6 ± 0.1 ***
line 2	4.3 ± 0.1	26.4 ± 2.4 **	13.6 ± 0.3 ***	5.5 ± 0.8 ***	10.3 ± 0.9 ***	38.8 ± 2.5 ***	1.1 ± 0.1 **	50.1 ± 1.9 **
line 3	5.2 ± 0.0 ***	35.7 ± 1.6 **	15.1 ± 0.4 ***	3.1 ± 1.0 ***	8.2 ± 1.1 ***	32.4 ± 1.8 ***	0.5 ± 0.3 **	40.9 ± 1.1 ***
line 4	4.1 ± 0.3	22.6 ± 0.6 ***	15.8 ± 1.5 ***	4.3 ± 2.5 **	7.5 ± 1.0 ***	44.9 ± 0.7 ***	0.7 ± 0.6 **	53.2 ± 1.3 **
line 5	5.4 ± 0.1 ***	30.2 ± 1.0 ***	13.2 ± 1.1 ***	3.8 ± 0.3 ***	6.5 ± 0.8 ***	37.3 ± 1.0 ***	0.7 ± 0.1 ***	44.5 ± 1.8 ***
line 6	4.5 ± 0.0 *	35.8 ± 4.1 ***	15.1 ± 0.2 ***	5.0 ± 0.7 ***	6.3 ± 0.4 ***	35.6 ± 3.7 ***	0.7 ± 0.3 ***	42.6 ± 4.1 **
line 7	4.1 ± 0.1	17.5 ± 0.8	17.3 ± 1.2 ***	3.9 ± 0.9 ***	6.3 ± 0.5 ***	50.2 ± 0.5 **	0.7 ± 0.0 ***	57.2 ± 0.6
line 8	4.3 ± 0.2 *	38.8 ± 3.5 ***	14.4 ± 0.9 ***	1.7 ± 0.3 ***	4.6 ± 0.3 ***	36.1 ± 2.8 ***	0 ± 0.3 ***	40.7 ± 2.6 ***
line 9	3.9 ± 0.2	24.6 ± 2.1 **	13.3 ± 0.5 ***	6.0 ± 0.5 ***	1.9 ± 0.2 ***	48.9 ± 1.4 **	1.2 ± 0.2 **	52.1 ± 1.4 **
line 10	4.5 ± 0.8	32.3 ± 2.8 ***	10.7 ± 1.3 ***	7.1 ± 1.1 ***	1.1 ± 0.2 ***	43.1 ± 2.2 **	1.5 ± 0.3 **	45.7 ± 1.8 ***
line 11	4.8 ± 0.3 *	22.9 ± 0.8 ***	13.9 ± 0.1 ***	7.0 ± 0.7 ***	0.7 ± 0.1 **	49.7 ± 0.7 **	0.9 ± 0.1 **	51.3 ± 0.6 ***
line 12	4.3 ± 0.2	28.4 ± 1.4 ***	16.0 ± 0.7 ***	3.0 ± 0.2 ***	0.4 ± 0.0	47.9 ± 0.7 **	0.0 ± 0.0 ***	48.3 ± 0.7 ***
line 13	4.4 ± 0.1	22.7 ± 2.8 *	17.4 ± 0.6 ***	1.9 ± 0.4 ***	0.4 ± 0.1	53.2 ± 2.0	0.0 ± 0.0 ***	53.6 ± 2.0
line 14	4.3 ± 0.3	24.3 ± 2.0 **	14.6 ± 1.1 ***	4.5 ± 0.9 ***	0.4 ± 0.0	50.9 ± 1.1*	1.1 ± 0.5 **	52.4 ± 1.6 **
line 15	4.0 ± 0.0 ***	26.3 ± 1.6 ***	14.5 ± 0.4 ***	4.0 ± 0.6 ***	0.4 ± 0.0	50.1 ± 0.9 *	0.7 ± 0.2 ***	51.2 ± 1.0 **
average of transgenic line	4.4 ± 0.4	27.8 ± 5.9	14.6 ± 1.7	4.4 ± 1.6	4.7 ± 4.5	43.5 ± 7.2	0.7 ± 0.4	48.9 ± 5.0

Three or four replicates of 30-seed samples were measured for wild-type and each transgenic line. All data are averages of three measurements ±SD. Fatty acid legend: 18:1 is oleic; 18:2 is linoleic; 18:3 is linolenic; 18:1OH is ricinoleic; 20:1OH is lesquerolic; and 20:2OH is auricolic acid. ^a^, total content of five common fatty acids: palmitic (16:0), palmitoleic (16:1), stearic (18:0), arachidic (20:0), and eicosenoic acids (20:1). Two-tailed Student’s *t*-test. * *p* < 0.05; ** *p* < 0.01; *** *p* < 0.001.

## Data Availability

Not applicable.

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
