# Peer review of "Genetic Engineering of Lesquerella with Increased Ricinoleic Acid Content in Seed Oil"

_plants, 2021, doi:10.3390/plants10061093_

Round 1

Reviewer 1 Report

The manuscript presents a genetic engineering of Lesquerella for improved production of ricinoleic acids in seeds, which is of interests for bioenergy and food industry. Overall, the manuscript is well-structured, and data are properly analyzed and presented. The main conclusions are supported by the data provided. The findings are novel, interesting and useful for future studies in engineering oil crops. The language is clear and easy to understand. I just have a few comments below.

  1. It is not clear why the authors specifically chose AtFAD3, CsFAE1, and CsFAD2 for the RNAi experiment. Some justifications in the introduction/results would be needed.
  2. The authors could use principal component analysis to compare the fatty acid profiles between the two transgenic lines and WT. The differences between the 2-dsRNA group and the 3-dsRNA group as discussed in line 187-195 should be supported by statistical analyses rather than the descriptive comparisons.
  3. Is there any other growth phenotypes recorded for these transgenic lines?
  4. Typo in “ACsFAE1” in line 258

Author Response

Our answers are in bold next to each Reviewer’s questions.

# 1 reviewer

Comments and Suggestions for Authors

The manuscript presents a genetic engineering of Lesquerella for improved production of ricinoleic acids in seeds, which is of interests for bioenergy and food industry. Overall, the manuscript is well-structured, and data are properly analyzed and presented. The main conclusions are supported by the data provided. The findings are novel, interesting and useful for future studies in engineering oil crops. The language is clear and easy to understand. I just have a few comments below.

  1. It is not clear why the authors specifically chose AtFAD3, CsFAE1, and CsFAD2 for the RNAi experiment. Some justifications in the introduction/results would be needed.

Answer: Justifications are added in Introduction (revision lines 128-132).

  1. The authors could use principal component analysis to compare the fatty acid profiles between the two transgenic lines and WT. The differences between the 2-dsRNA group and the 3-dsRNA group as discussed in line 187-195 should be supported by statistical analyses rather than the descriptive comparisons.

Answer: Principal Component Analysis (PCA), is a dimensionality-reduction method that is often used to reduce the dimensionality of large data sets by transforming a large set of variables into a smaller one. Reducing the number of variables of a data set naturally comes at the expense of accuracy. In our case, we do not have large data set, we provide descriptive comparisons which would be more accurate than PCA. Besides, our description is also simple and straight forward for understanding by audience of Plants.

  1. Is there any other growth phenotypes recorded for these transgenic lines?

Answer: A sentence,We did not observe any change of growth phenotype for all transgenic lesquerella lines.” is added in revision lines 209-210.

  1. Typo in “ACsFAE1” in line 258

Answer: the “ACsFAE1” is corrected to be “CsFAE1” in revision line 278.  

My answers are in bold next to each Reviewer’s questions.

# 1 reviewer

Comments and Suggestions for Authors

The manuscript presents a genetic engineering of Lesquerella for improved production of ricinoleic acids in seeds, which is of interests for bioenergy and food industry. Overall, the manuscript is well-structured, and data are properly analyzed and presented. The main conclusions are supported by the data provided. The findings are novel, interesting and useful for future studies in engineering oil crops. The language is clear and easy to understand. I just have a few comments below.

  1. It is not clear why the authors specifically chose AtFAD3, CsFAE1, and CsFAD2 for the RNAi experiment. Some justifications in the introduction/results would be needed.

Answer: Justifications are added in Introduction (revision lines 128-132).

  1. The authors could use principal component analysis to compare the fatty acid profiles between the two transgenic lines and WT. The differences between the 2-dsRNA group and the 3-dsRNA group as discussed in line 187-195 should be supported by statistical analyses rather than the descriptive comparisons.

Answer: Principal Component Analysis (PCA), is a dimensionality-reduction method that is often used to reduce the dimensionality of large data sets by transforming a large set of variables into a smaller one. Reducing the number of variables of a data set naturally comes at the expense of accuracy. In our case, we do not have large data set, we provide descriptive comparisons which would be more accurate than PCA. Besides, our description is also simple and straight forward for understanding by audience of Plants.

  1. Is there any other growth phenotypes recorded for these transgenic lines?

Answer: A sentence,We did not observe any change of growth phenotype for all transgenic lesquerella lines.” is added in revision lines 209-210.

  1. Typo in “ACsFAE1” in line 258

Answer: the “ACsFAE1” is corrected to be “CsFAE1” in revision line 278.  

Reviewer 2 Report

The author made efforts to decipher complex mechanisms of oil biosynthesis in lesquerella and showed the potential of biotechnology to tailor the oil functionality of this industrial oil feedstock. This work might be of great importance however, It requires substantial improvements. My comments are below.

----------------------------------------

I have read the entire manuscript and my initial comment is that manuscript is poorly written and needs to reflect the significant finding or novelty. 

I have significant concerns about the grammar and vocabulary of the manuscript; therefore, the improvement of the language is highly needed. 

Overall, the amount of work is very less and the data are short and just does not meet the criteria of publication in current form (need more figures and supplementary data)

Author Response

Our answers to the questions from #2 Reviewer:

#2 reviewer

Comments and Suggestions for Authors

The author made efforts to decipher complex mechanisms of oil biosynthesis in lesquerella and showed the potential of biotechnology to tailor the oil functionality of this industrial oil feedstock. This work might be of great importance however, it requires substantial improvements. My comments are below.

  1. I have read the entire manuscript and my initial comment is that manuscript is poorly written and needs to reflect the significant finding or novelty.

Answer: To indicate ‘novelty’, we added “the first time” in revision line 379. We also indicate the significances of this research in revision lines, 25-27, 131-134, 377-378.

  1. I have significant concerns about the grammar and vocabulary of the manuscript; therefore, the improvement of the language is highly needed.

Answer: the manuscript revision has been proof read.

Overall, the amount of work is very less and the data are short and just does not meet the criteria of publication in current form (need more figures and supplementary data)

Answer: we do not agree with the Reviewer’s opinion that “the amount of work is very less”. The amount of work for this research took more than four years to complete, due to the difficulties of lesquerella genetic transformation and seed production described previously [#14 citation of manuscript]. The Figures and Tables provided in this manuscript are definitive and sufficient results, which allow us to discuss findings and draw conclusions.

#2 reviewer

Comments and Suggestions for Authors

The author made efforts to decipher complex mechanisms of oil biosynthesis in lesquerella and showed the potential of biotechnology to tailor the oil functionality of this industrial oil feedstock. This work might be of great importance however, it requires substantial improvements. My comments are below.

  1. I have read the entire manuscript and my initial comment is that manuscript is poorly written and needs to reflect the significant finding or novelty.

Answer: To indicate ‘novelty’, we added “the first time” in revision line 379. We also indicate the significances of this research in revision lines, 25-27, 131-134, 377-378.

  1. I have significant concerns about the grammar and vocabulary of the manuscript; therefore, the improvement of the language is highly needed.

Answer: the manuscript revision has been proof read.

Overall, the amount of work is very less and the data are short and just does not meet the criteria of publication in current form (need more figures and supplementary data)

Answer: we do not agree with the Reviewer’s opinion that “the amount of work is very less”. The amount of work for this research took more than four years to complete, due to the difficulties of lesquerella genetic transformation and seed production described previously [#14 citation of manuscript]. The Figures and Tables provided in this manuscript are definitive and sufficient results, which allow us to discuss findings and draw conclusions.

#2 reviewer

Comments and Suggestions for Authors

The author made efforts to decipher complex mechanisms of oil biosynthesis in lesquerella and showed the potential of biotechnology to tailor the oil functionality of this industrial oil feedstock. This work might be of great importance however, it requires substantial improvements. My comments are below.

  1. I have read the entire manuscript and my initial comment is that manuscript is poorly written and needs to reflect the significant finding or novelty.

Answer: To indicate ‘novelty’, we added “the first time” in revision line 379. We also indicate the significances of this research in revision lines, 25-27, 131-134, 377-378.

  1. I have significant concerns about the grammar and vocabulary of the manuscript; therefore, the improvement of the language is highly needed.

Answer: the manuscript revision has been proof read.

Overall, the amount of work is very less and the data are short and just does not meet the criteria of publication in current form (need more figures and supplementary data)

Answer: we do not agree with the Reviewer’s opinion that “the amount of work is very less”. The amount of work for this research took more than four years to complete, due to the difficulties of lesquerella genetic transformation and seed production described previously [#14 citation of manuscript]. The Figures and Tables provided in this manuscript are definitive and sufficient results, which allow us to discuss findings and draw conclusions.

#2 reviewer

Comments and Suggestions for Authors

The author made efforts to decipher complex mechanisms of oil biosynthesis in lesquerella and showed the potential of biotechnology to tailor the oil functionality of this industrial oil feedstock. This work might be of great importance however, it requires substantial improvements. My comments are below.

  1. I have read the entire manuscript and my initial comment is that manuscript is poorly written and needs to reflect the significant finding or novelty.

Answer: To indicate ‘novelty’, we added “the first time” in revision line 379. We also indicate the significances of this research in revision lines, 25-27, 131-134, 377-378.

  1. I have significant concerns about the grammar and vocabulary of the manuscript; therefore, the improvement of the language is highly needed.

Answer: the manuscript revision has been proof read.

Overall, the amount of work is very less and the data are short and just does not meet the criteria of publication in current form (need more figures and supplementary data)

Answer: we do not agree with the Reviewer’s opinion that “the amount of work is very less”. The amount of work for this research took more than four years to complete, due to the difficulties of lesquerella genetic transformation and seed production described previously [#14 citation of manuscript]. The Figures and Tables provided in this manuscript are definitive and sufficient results, which allow us to discuss findings and draw conclusions.

#2 reviewer

Comments and Suggestions for Authors

The author made efforts to decipher complex mechanisms of oil biosynthesis in lesquerella and showed the potential of biotechnology to tailor the oil functionality of this industrial oil feedstock. This work might be of great importance however, it requires substantial improvements. My comments are below.

  1. I have read the entire manuscript and my initial comment is that manuscript is poorly written and needs to reflect the significant finding or novelty.

Answer: To indicate ‘novelty’, we added “the first time” in revision line 379. We also indicate the significances of this research in revision lines, 25-27, 131-134, 377-378.

  1. I have significant concerns about the grammar and vocabulary of the manuscript; therefore, the improvement of the language is highly needed.

Answer: the manuscript revision has been proof read.

Overall, the amount of work is very less and the data are short and just does not meet the criteria of publication in current form (need more figures and supplementary data)

Answer: we do not agree with the Reviewer’s opinion that “the amount of work is very less”. The amount of work for this research took more than four years to complete, due to the difficulties of lesquerella genetic transformation and seed production described previously [#14 citation of manuscript]. The Figures and Tables provided in this manuscript are definitive and sufficient results, which allow us to discuss findings and draw conclusions.

Reviewer 3 Report

Comments file attached.

Author Response

Our answers to #3 Reviewer are in bold below:

The manuscript entitled “Genetic Engineering of Lesquerella with Increased Ricinoleic Acid Content in Seed Oil” by Chen et al developing a genetically engineered Lesquerella in order to increase the production of castor oil using RNAi technology is interesting and  worth publishing.

Suggestions and corrections

  1. The author has excelled in explaining the facts identified in this report, but has neglected to relate the results to pathways involving ricinoleic acid and lesquerolic acid. How they are increasing/decreasing in transgenic lines?

Answer: The increasing of ricinoleic acid and decreasing lesquerolic acid in transgenic lines are caused by silencing PfKCS18 gene involved in elongation of ricinoleic acid to lesquerolic acid in the pathway, discussed in revision lines 246-249.    

  1. The author didn’t not discuss the level of oleic and linolenic with suitable reference?

Answer: We add a refence [11] in revision line 205-207. The levels of oleic and linolenic from this work are listed in Table 1, Table 2 and Figure 2. We discussed relationships among various fatty acid composition changes (Table S4), including oleic (18:1), linoleic (18:2) and linolenic (18:3) acids, in revision lines 244-249, 288-292, 307-310.

  1. The author has not explained the decreased ricinoleic acid production in T2 compared to T1?

Answer: the decreased ricinoleic acid production in T2 compared to T1 is explained by “ the competence of CsFAE1 RNAi was not fully transmitted to the next generation” (revision line 256-258), and the fading of the silencing is discussed in revision line 258-275.

  1. Why did you select this variety (WCL-LY2) for this study and their importance?

Answer: there is no particular reason for which line to be used in this study. Dr. Dierig is a lesquerella breeder, WCL-LY2 lesquerella line was available when this research started.

  1. The authors should clearly summarize the main finding of the research.

Answer: a summary paragraph is added in revision line 369-380.  

  1. The results should be interpreted meaningfully with the supporting references.

Answer: In Discussion section 3.1., the results are interpreted meaningfully with supporting references, including 56, 57, 58, 59, 60, 62. In section 3.2., results are discussed based on solid evidences cited, 15, 17, 19, 24, 25, 27, 33, 35, 37, 40, 41, 44, 45, 46, 50, 52, 61, 63, 64, 65, 66, 67, 68.

  1. Please increase the resolution of the Figure 1. Also, Tables 1, 2 were given as picture format, text is blurred. Please check it once and improve the quality

Answer: Authors uploaded the original Figure 1 and Figure 2 and requested the editorial office of Plants to improve the resolution. Resolution of Table 1 and Table 2 are improved by typing the content in the manuscript.

  1. Scientific names should be in italics, throughout the manuscript.

Answer: scientific names are in italics throughout the manuscript. Authors note here that Arabidopsis, camelina and lesquerella are common names so they are not in italics. 

  1. Lines 402-404, write transformation protocol in brief. May be not necessary to refer

the cross reference for the readers.

Answer: Authors wrote the transformation protocol in brief in revision line 435-455.

  1. Overall, this manuscript's language is satisfactory. However, I would recommend thorough proofreading to correct any grammatical or usage errors.

Answer: the manuscript revision has been proof read.

As a result, the overall recommendation is to address the minor suggestions in the manuscript with moderate revisions that meet the above-mentioned requirements

Round 2

Reviewer 2 Report

I can see that the authors did many improvements, so at this point, I am endorsing it for acceptance.